# Comparison between Cultivated Oral Mucosa and Ocular Surface Epithelia for COMET Patients Follow-Up

**DOI:** 10.3390/ijms241411522

**Published:** 2023-07-15

**Authors:** Eustachio Attico, Giulia Galaverni, Andrea Torello, Elisa Bianchi, Susanna Bonacorsi, Lorena Losi, Rossella Manfredini, Alessandro Lambiase, Paolo Rama, Graziella Pellegrini

**Affiliations:** 1Centre for Regenerative Medicine “Stefano Ferrari”, University of Modena and Reggio Emilia, 41125 Modena, Italy; giulia.galaverni@unimore.it (G.G.); a.torello@holostem.com (A.T.); elisa.bianchi@unimore.it (E.B.); susanna.bonacorsi@unimore.it (S.B.); rossella.manfredini@unimore.it (R.M.); graziella.pellegrini@unimore.it (G.P.); 2Holostem Terapie Avanzate s.r.l., 41125 Modena, Italy; 3Unit of Pathology, Department of Life Sciences, University of Modena and Reggio Emilia, 41124 Modena, Italy; lorena.losi@unimore.it; 4Department of Sense Organs, Sapienza University of Rome, 00169 Rome, Italy; alessandro.lambiase@uniroma1.it; 5SC Ophathalmology, IRCCS Policlinico San Matteo Foundation, 27100 Pavia, Italy; p.rama@smatteo.pv.it

**Keywords:** aniridia, biomarker, COMET, cornea, LSCD, neovascularization, oral mucosa, PITX2, ocular surface

## Abstract

Total bilateral Limbal Stem Cell Deficiency is a pathologic condition of the ocular surface due to the loss of corneal stem cells. Cultivated oral mucosa epithelial transplantation (COMET) is the only autologous successful treatment for this pathology in clinical application, although abnormal peripheric corneal vascularization often occurs. Properly characterizing the regenerated ocular surface is needed for a reliable follow-up. So far, the univocal identification of transplanted oral mucosa has been challenging. Previously proposed markers were shown to be co-expressed by different ocular surface epithelia in a homeostatic or perturbated environment. In this study, we compared the transcriptome profile of human oral mucosa, limbal and conjunctival cultured holoclones, identifying Paired Like Homeodomain 2 (*PITX2*) as a new marker that univocally distinguishes the transplanted oral tissue from the other epithelia. We validated PITX2 at RNA and protein levels to investigate 10-year follow-up corneal samples derived from a COMET-treated aniridic patient. Moreover, we found novel angiogenesis-related factors that were differentially expressed in the three epithelia and instrumental in explaining the neovascularization in COMET-treated patients. These results will support the follow-up analysis of patients transplanted with oral mucosa and provide new tools to understand the regeneration mechanism of transplanted corneas.

## 1. Introduction

Limbal stem cells (LSCs) play a major role in ocular surface homeostasis and are responsible for corneal regeneration, achieved through centripetal cell migration and differentiation from the limbus to the central cornea [1]. As a result of acquired (chemical/thermal burns) or genetic conditions, these cells can be partially or entirely lost, resulting in a pathology known as Limbal Stem Cell Deficiency (LSCD), where corneal opacity and conjunctivalization lead to visual impairment and blood vessels migration, otherwise absent over the central cornea [2,3]. With a total prevalence of 1–5:10,000 individuals, LSCD received the orphan drug designation in 2008 (Orpha:171673) from the Committee for Orphan Medicinal Products [4,5,6].

Over the years, several treatments have been proposed to treat unilateral or partial bilateral LSCD, including conjunctival limbal allograft (CLAL) [7], conjunctival limbal autograft (CLAU) [8], cultured limbal epithelial transplantation (CLET) [9] or simple limbal epithelial transplantation (SLET) [10].

The CLET treatment, consisting of the transplantation of autologous limbal cells cultured on fibrin and clinical-grade 3T3-J2 feeder cells, obtained conditional approval from the European Medicines Agency (EMA) [5,11] under the name of Holoclar^®^ and resulted in long-term effectiveness in 85.19% of patients with partial LSCD [12,13]. However, none of these therapeutic options are feasible in the case of total bilateral LSCD, where the absence of LSCs available for ex vivo expansion implies the need for an alternative cell source.

Cultivated oral mucosa epithelial transplantation (COMET) is an alternative procedure, using oral mucosa cells in place of limbal, that has proven to be well tolerated and effective in patients affected by total bilateral LSCD [14]. However, after several years of application of this clinical procedure by different research groups, little is known about the mechanism of action that sustains corneal regeneration [15]. The two main hypotheses underlying the restoration of corneal transparency and visual acuity are the engraftment of oral mucosal transplanted cells with subsequent replacement of the epithelium (“engraftment” hypothesis) or the stimulation of few residual not detectable LSCs to proliferate and restore the ocular surface (“stimulation” hypothesis) [16].

Different studies in animal models supported both hypotheses. The transplantation of oral tissue from Green Fluorescence Protein (GFP)-tagged rats onto the ocular surface of nude rat LSCD models showed maintenance of the oral epithelium [17], while autologous oral mucosa cells transplanted onto rabbit LSCD models revealed the stimulation of residual LSCs after repeated wounding of the ocular surface [18]. However, the various technical procedures and the different physiologies of animal models limit the significance of these studies.

A targeted examination of the follow-up specimens derived from COMET patients can help to understand the biological mechanism sustaining the repair and gain new insights into oral mucosal cells’ plasticity to adapt to the corneal environment and functions and enable the analysis and stratification of successes and failures. To this aim, specific markers are needed to distinguish the oral mucosa, corneal and conjunctival epithelia.

Within this scope, cytokeratins have been widely employed. However, several concerns remain about their expression change in specific conditions, such as in a proliferating/wound-healing state [19,20,21]. Our previous work suggested SOX2 as a univocal marker to distinguish the oral mucosa from corneal and conjunctival cells based on an unbiased gene array transcriptome profile analysis of single cells derived from different regions [20].

Nevertheless, a broad panel of markers is needed to analyze the specimens derived from COMET patients, providing a more consistent picture of the clinical condition. Thus, the original analysis is augmented by additional marker investigations.

A complementary issue is peripheral corneal neoangiogenesis, especially in the limbal area after COMET treatment [14,22]. Indeed, the oral mucosa (highly vascularized in situ) leads to the formation of neovessels in only the limbo-conjunctival area, not in the central cornea, when transplanted over the ocular surface. The vascularization of the central cornea impairs the transparency required for visual acuity and alters the corneal microenvironment with a degeneration of tissue integrity.

In this work, a particular focus was given to vascularization-associated factors to shed more light on this phenomenon.

In summary, this work will support the characterization of the ocular surface epithelium in patients undergoing COMET, thus understanding the biological mechanism that drives corneal repair and peripheral neovascularization.

## 2. Results

### 2.1. Gene Expression Profiling of Holoclones from Oral Mucosa, Limbus and Conjunctiva

The functional analysis of the oral mucosa epithelium for COMET patient characterization was performed by microarray analysis of stem-cell-containing clones isolated by clonal analysis, as previously described [20]. Briefly, 32 holoclones (clones with less than 5% of abortive colonies, as defined in paragraph 5.5) were subcultured from oral mucosa (n = 15), corneal (n = 8) and conjunctival (n = 9) tissues and processed for RNA extraction and microarray analysis (Figure 1).

### 2.2. PITX2 mRNA Is Overexpressed in Oral Mucosa Compared to Ocular Surface Tissues

To identify unique tissue markers, we focused on the most differentially expressed transcripts in the pairwise comparisons between the oral mucosa and the ocular surface epithelia (Spreadsheets S1–S3 in [20]).

Comparing the oral mucosa to the ocular surface epithelia, the most upregulated transcript was Paired Like Homeodomain 2 (*PITX2*, FC = 65.80 and FC = 77.28 in oral mucosa vs. limbus and conjunctiva, respectively) (Figure 2A).

Real-time PCR analyses confirmed that *PITX2* was upregulated in the stem-cell-containing holoclones of oral mucosa compared to holoclones from limbus and conjunctiva (Figure 2B).

Moreover, we tested oral mucosa, limbus and conjunctiva bulk transcriptomes by real-time PCR, intending to evaluate the expression of the *PITX2* transcript in holoclones and primary cultures. The data confirmed upregulation only in the oral mucosa (Figure 2C).

The downregulation of *PITX2* was also demonstrated by comparing oral mucosa holoclones to those derived from other epithelia, such as the limbus, conjunctiva, epidermis and urethra (Figure 2D).

We further analyzed *PITX2* expression in oral mucosa meroclones, which are the progeny of the transient amplifying cells (TA-cells) and retain a lower clonogenicity if compared to holoclones [23,24]. The results showed that *PITX2* mRNA did not correlate with the clonogenicity of the clones (Figure 2E). Moreover, we also analyzed the expression of the transcript over consecutive passages of the oral mucosa, which was cultivated until stem cell exhaustion and senescence. *PITX2* was expressed in young, medium and old passages of two different strains and showed a statistically significant upregulation in the youngest passage compared to the other passages of the lifespan (Figure 2F). This evidence may suggest a correlation between *PITX2* expression levels and the high proliferative potential of the cells, which is peculiar of the youngest passages.

### 2.3. Analysis of PITX2 Isoforms

Since transcriptional factor *PITX2* has three major different isoforms (*PITX2A*, *PITX2B* and *PITX2C*) [25], we evaluated which isoforms were expressed in the oral mucosa. Specific primers were designed, and an oral mucosa holoclone was analyzed by real-time PCR. The results highlighted high levels of the *PITX2C* isoform and low levels of *PITX2A* and *PITX2B* (Figure 2G).

The same analysis was also conducted on early passages of the primary oral mucosa, limbus and conjunctiva. While the upregulation of *PITX2C* was confirmed, we observed a high expression of the *PITX2B* isoform (Figure 2H). Instead, *PITX2A* was expressed at low levels. According to precedent findings, all *PITX2* isoforms were strongly downregulated in the limbus and conjunctiva compared to the oral mucosa (Figure 2H).

### 2.4. Validation of the Results by In Situ Hybridization (ISH)

In order to validate the microarray results in in vivo samples, oral mucosa and corneal donor specimens were tested through an ISH assay using a pan-*PITX2* probe, corresponding to a common target sequence for all isoforms. In the oral mucosa epithelium, the *PITX2* transcript was specifically detected in basal and suprabasal keratinocytes up to the granular layer, resulting in the absence of the central cornea, limbus and conjunctiva (Figure 3A). Coherently with the literature, the corneal endothelium was *PITX2*-positive [26,27].

### 2.5. Validation of the Results at Protein Level

To evaluate PITX2 as a putative tissue marker at a protein level, we performed indirect immunofluorescence staining on OCT frozen samples of in vivo limbo-cornea and oral mucosa samples. The PITX2 protein showed nuclear expression in the basal and suprabasal tiers of the oral mucosa epithelium up to the granular layer. In contrast, we could not detect any expression in the central cornea, limbus or conjunctiva (Figure 3B). The same results were obtained through immunohistochemical analysis on FFPE sections (Figure 3C).

### 2.6. Phenotypic Characterization of the Patient after COMET

Once PITX2 was validated as a marker for the oral mucosa epithelium, we phenotypically characterized three corneal buttons obtained from an aniridic patient who underwent consecutive interventions of penetrating keratoplasty (PK) after a COMET procedure. At admission, the patient’s left eye was covered by a conjunctival pannus, and blood vessels migrated to the central cornea. During the COMET procedure, the fibrovascular tissue was removed, and the oral mucosal cells cultured over a fibrin scaffold were applied to the ocular surface. After the treatment, the epithelium was intact, and the blood vessels partially regressed, although the stroma remained opaque. PKs were performed at 1, 3 and 10 years after COMET to restore corneal transparency. The three corneal buttons, previously analyzed for different markers [20], revealed nuclear staining in the regenerated epithelium by IHC analysis for PITX2. The marker was more expressed in the 3-yy specimen, being the most stratified (Figure 4A(c,d)). However, in this sample there were other areas found to be negative (Figure 4A(e,f)).

In the samples obtained from the COMET after 1 and 10 years, the epithelium was thinner, and PITX2 resulted positive in only a few areas (Figure 4A(a,g)) and was absent in the rest (Figure 4A(b,h)). Detecting goblet cells in the 3-yy specimen confirmed the coexistence with conjunctival tissue (Figure 4A(f)) [20].

Overall, these findings revealed the presence of oral mucosa tissue up to 10 years after the surgery and several PKs, highlighting the long-term maintenance of the transplanted cells, even though the highest abundance of the PITX2 marker was detected in the stratified epithelium of the 3-yy follow-up specimen (Figure 4B).

### 2.7. Angiogenic and Antiangiogenic Comparison between Oral Mucosa, Limbus and Conjunctiva

One of the significant postoperative problems in COMET patients is the invasion of blood vessels over the graft, which causes pain and visual acuity reduction, leading to total or partial failure of the procedure [14,22]. To date, there is no evidence of why only some patients develop such a phenomenon.

To unravel this issue, starting from microarray data predicting an inhibition of angiogenesis in the oral mucosa versus conjunctiva (Figure 5A), we focused on the genes differentially expressed in the oral mucosa in comparison with the limbus and conjunctiva and coding for factors involved in angiogenesis. We found nine transcripts codifying for proangiogenic (*AGR2*, *CRYAB*, *EREG*, *S100A4* and *JAM-3*) or antiangiogenic (*COL4A1*, *COL4A2*, *IL1RN* and *TIMP2*) proteins (Figure 5B).

#### 2.7.1. Proangiogenic Factors

*AGR2* and *CRYAB* transcripts were upregulated in the conjunctiva and in the limbus compared to the oral mucosa and have been related to neoangiogenesis. The extracellular AGR2 binds VEGF and FGF-2, increasing their proangiogenic activity [28,29], while CRYAB increases choroidal neoangiogenesis through the VEGF signaling pathway, acting as a chaperone for VEGF-A [30,31].

Comparing the conjunctiva and oral mucosa, we found two more transcripts encoding for proangiogenic factors: *EREG* and *S100A4*. The transcript *EREG* is translated to epiregulin (EPR), a member of the EGF family, which is secreted via exosomes and has a key role in promoting angiogenesis through the upregulation of VEGF-A and FGF-2 [32]. At the same time, the metastasis-associated protein S100A4 induces angiogenesis by binding Annexin II and accelerates plasmin formation [33,34].

Moreover, the *JAM-3* transcript, encoding the JAM-C proangiogenic peptide (often targeted in antitumoral therapies) was upregulated in the oral mucosa vs. limbus [35]. Although more studies are needed to explore this hypothesis, the expression of JAM-C could play a role in corneal peripheral vascularization post-COMET.

#### 2.7.2. Antiangiogenic Factors

Conversely, the antiangiogenic transcripts *COL4A1*, *COL4A2* and *IL1RN* were upregulated in the oral mucosa vs. conjunctiva. The alpha chains 1 and 2 of collagen IV (*COL4A1*, *COL4A2*) have been reported to produce C-terminal-derived peptides (called arresten and canstatin, respectively) endowed with antiangiogenic properties [36,37]. In addition, IL1RN (or IL1RA) decreases the inflammatory environment by downregulating IL-1β, IL-6 and vascular adhesion molecule VCAM-1, and suppresses corneal neovascularization [38]. The ability of these factors to limit conjunctival blood vessels’ ingrowth could explain the avascularity of the oral mucosa in the central ocular surface after COMET.

Surprisingly, the antiangiogenic metalloproteinase inhibitor *TIMP2* was downregulated in the oral mucosa vs. conjunctiva. TIMP2 was shown to be released by the amniotic membrane in culture, significantly suppressing the corneal neovascularization induced by FGF-2 [39].

A deep analysis of the interplay of these factors would explain the blood vessels’ outgrowth over the periphery of the transplanted oral mucosa or their absence in the central part.

## 3. Discussion

Since 2004, the oral mucosa epithelium has proven to be an optimal alternative for treating total bilateral LSCD patients with a procedure named COMET [14,22]. Using an autologous source of epithelial stem cells overcomes the problems related to autologous stem cell shortage and allogenic transplantation, rejection, and lifelong immunosuppression.

One of the distinctive features of stem cell presence in the limbus is corneal avascularity. The central corneal lacks capillaries, and many factors are involved in the so-called “angiogenic privilege”. However, many alterations can affect this homeostasis, promoting corneal neovascularization [40]. Pathological conditions such as viral infections, primary or secondary inflammations, degeneration of the limbus due to congenital pathologies (e.g., congenital aniridia), traumas, hypoxia and neoplasia can trigger corneal neoangiogenesis. Most of these conditions can also be listed among the causes of LSCD, and neovascularization is one of the consequences that exacerbates its symptoms. The presence of the conjunctival pannus highlights the LSCD, which is associated with the invasion of blood vessels, thus participating in a positive feedback loop. In this context, the lack of the antiangiogenic factor thrombospondin-1 (TSP-1), normally expressed by corneal cells, is likely to play a major role [41]. A bilateral LSCD patient undergoing COMET treatment is characterized by this condition (pre-operation neovascularization).

Neovascularization can also arise in the transplanted corneas after the COMET procedure (post-operation neovascularization). This phenomenon is facilitated by post-operative inflammation and occurs mainly in the peripheric corneal region [14,22]. In normal conditions, the production of TSP-1 by keratocytes may limit vascularization to this area [22,41] while, after a corneal wound, a high presence of ANG2, especially released by keratocytes, helps in the formation of neovessels [42]. Finally, the presence of an oral mucosa could also trigger angiogenesis because, in vivo, this tissue requires blood supply for its maintenance.

When the balance between pro- and antiangiogenic factors is altered due to physiological or pathological conditions, it results in a decrease in or formation of new blood vessels. This equilibrium has been widely studied on the ocular surface, although an elucidation of all the mechanisms and molecules involved is still far away [40]. In this process, the tear film is revealed to play a key role, containing several proangiogenic substances, such as IL-6, IL-8, and VEGF [43].

Altogether, the absence of sFlt-1, TIMP-3, and TSP-1 has been described in COMET specimens compared to normal corneas, suggesting their involvement in “angiogenic privilege” and therefore in the peripheric neovascularization observed in COMET patients [44]. Moreover, the FGF2 factor was also reported to participate in this process [45].

We identified nine factors related to angiogenetic processes in the presented unbiased comparison between oral mucosa and ocular surface progenitor cells. Five were associated with proangiogenic capacities (AGR2, CRYAB, EPR, S100A4 and JAM-C), while four were reported to have an antiangiogenic role (COL4A1, COL4A2, IL1RN and TIMP2).

Worthy of note, the upregulation of the proangiogenic JAM-C in the oral mucosa (compared to the limbus) could be an exciting clue to be investigated in post-COMET peripheral neoangiogenesis. This factor should be studied by loss- or gain-of-function experiments to correlate its expression to the prognosis.

Moreover, the transcripts of the potential antiangiogenic peptides COL4A1, COL4A2 and IL1RN were found to be upregulated in the oral mucosa compared to the conjunctiva, highlighting the capacity of the former tissue to stop the progression of conjunctival blood vessels towards the central ocular surface in COMET patients. Further studies could confirm this role.

The effective mechanism of ocular surface regeneration after oral mucosa transplantation is still unclear. The literature confirms that the engraftment of the oral mucosa tissue plays an essential role in the short term after the transplantation; however, follow-up data reveal its presence up to 10 years later [20]. Together with these insights supporting the engraftment of oral mucosal cells, their possible role in stimulating some residual limbal stem cells should also be considered, as well as a mixed pattern of the two regeneration mechanisms (Figure 6). Indeed, corneal tissue was detected on COMET-transplanted eyes, although was presumed to be completely depleted in patients suffering from total bilateral LSCD [15,46,47]. Moreover, donor corneal cells were also revealed in total bilateral LSCD patients treated with limbal allografts [48,49]. Such host limbal stem cells may be too few to duplicate in a pathologic contest, and they could regenerate the corneal tissue when triggered by exogenous stimuli [16].

The means by which the oral tissue could regenerate the ocular surface are different: paracrine signaling, cytokine stimulation, exosome delivery, growth factor release, direct contact communications and others. In recent investigations, exosomes and extracellular vesicles (EVs) released from the oral mucosa showed great regenerative potential [50,51]. In addition, the proangiogenic factor EPR was found in oral mucosa exosomes [32]. Considering these findings, oral mucosa EVs should be studied in the post-COMET environment, also in relation to peripheral neovascularization. Moreover, depicting the factors released by the oral mucosa could result in a pharmacological therapeutic option for LSCD patients.

Several studies assume that the presence of corneal cells observed in patients after COMET is due to a change in the phenotype of oral cells into corneal one, a phenomenon called transdifferentiation [22,52]. However, it has been reported that epithelial cells transplanted in ectopic districts maintain their original phenotype, making the transdifferentiation hypothesis unlikely [53,54,55].

In the literature, different markers were adopted to identify the three possible epithelia on the ocular surface after COMET, namely, the oral mucosa, the cornea, or the conjunctiva [46,47,56,57,58]. Cytokeratins have been widely used to characterize these tissues; for example, K3 and K12 identify the cornea and K13 and K19 mainly identify the conjunctiva. However, the oral mucosa shares the expression of some of these markers (K3 and K13) with the ocular tissues [59]. Worthy of note, the expression of cytokeratins differs considering the condition of the tissue. Indeed, these markers can be activated as a result of wound healing or inflammation [19,20,21], and their expression can change considerably due to pathologic processes, including genetic conditions such as aniridia [60,61].

Recently, our group has proposed the SOX2 transcription factor as a marker that univocally distinguishes the presence of oral mucosa tissue on the ocular surface of patients who underwent COMET and does not change its expression during wound healing [20]. This study found a new marker to distinguish the oral tissue from the ocular surface. In the unbiased comparison among progenitor cells derived from the cornea, conjunctiva, and oral mucosa, we found that *PITX2* was the most differentially expressed gene. This finding was validated on mRNA and protein levels, both in vitro and in vivo.

PITX2 is related to the development of specific tissues and organs, including the cornea [62]. It has a pivotal role in the determination of left-right asymmetry in vertebrates [63,64] and in the morphogenesis of the pituitary gland [65], teeth [66], skeletal muscle [67], heart [68], brain [69], etc. Recently, a PITX2-SOX2 interaction was described in the progenitor oral/dental epithelial cell signaling center specification during odontogenesis [63]. PITX2 was also associated with corneal development and was reported as necessary for establishing corneal angiogenic privilege by upregulating AP-2β and other genes [62,70]. However, in the adult cornea, PITX2 expression is confined only to endothelial cells [26,27]. Moreover, mutations in PITX2 were found in patients with defects in the eye anterior chamber, such as Rieger syndrome [71]. Finally, the overexpression of this peptide was adopted as a tumorigenic hallmark in different districts, such as esophageal squamous cell carcinoma (ESCC) [72] and prostate [25], colorectal [73], ovarian [74] and thyroid cancers [75].

PITX2 can be found in at least three isoforms (namely PITX2-A, PITX2-B and PITX2-C) that correlate to diverse roles depending on their expression [76,77].

This paper explored the different isoforms expressed by cultured oral mucosa, revealing that *PITX2-B* and *PITX2-C* transcripts were the most represented.

In vivo, we observed that *PITX2* mRNA, detected by an ISH assay, was mainly expressed from the basal up to granular layer. This finding was also confirmed at a protein level by IF and IHC assays. Indeed, in culture, we did not identify a strong correlation between oral mucosa stem cells (holoclones) and more differentiated cells (meroclones). Nevertheless, when we investigated its expression within serial passages of cultured oral mucosa, we detected a significative higher expression in the youngest passage, suggesting a possible involvement in the proliferative potential.

Finally, we used the PITX2 marker to analyze three corneal buttons obtained from the same aniridic patient who underwent repeated penetrating keratoplasties 1, 3 and 10 years after COMET procedure. The same samples were previously analyzed for their expression of canonical markers (i.e., K3, K12, K13, Alcian Blue/PAS and PAX6), showing the presence of goblet cells, and thus conjunctiva [20]. Moreover, the expression of the SOX2 marker highlighted the presence of the oral mucosa throughout different time points, especially after 3 years, when the epithelium was more stratified and morphologically closer to the in vivo oral mucosa. Herein, these observations were implemented with the analysis of PITX2, whose nuclear positivity in all three corneal button specimens confirmed the persistence of oral mucosa up to 10 years. Consistent with previous observations, the 3-year corneal button showed the highest positivity, probably due to the high epithelial stratification of the ocular surface at that time point.

The identification of markers such as SOX2 and PITX2 that unequivocally identify the oral mucosa in the follow-up samples of COMET patients will be instrumental in extending this analysis to a larger cohort of COMET patients (comprising different LSCD etiologies, oral mucosa culture methods and neovascularization stages), shedding light on the possible regenerative mechanism of this treatment.

## 4. Materials and Methods

### 4.1. Patients and Specimens

Specimens were obtained in accordance with the tenets of the Declaration of Helsinki; donors provided informed consent for biopsies. Permission was also obtained for samples taken from organ donors. Corneal and conjunctival specimens were obtained from the ocular surface of donors or cadavers, while small oral mucosal biopsies were collected from the inner cheek or inferior labial of patients undergoing oral mucosa transplantation for urethral stricture treatment [78,79].

### 4.2. COMET Transplantation

Eleven years before the COMET procedure, a 41-year-old woman suffering from total bilateral LSCD and glaucoma due to congenital aniridia underwent an anterior lamellar keratoplasty in her left eye, which failed due to superficial neovascularization caused by limbal deficiency. Under para/retrobulbar anesthesia, the conjunctiva was released a few millimeters outside the limbus, exposing the sclera, and the fibrovascular corneal pannus was removed. After this step, the oral mucosa cultured on a fibrin sheet was transferred to the corneal area; the excess of the fibrin was trimmed and the edges were covered with the conjunctiva, applying 2 or 3 stitches of vicryl or silk 8/0.

### 4.3. Cell Cultures

Oral mucosal, conjunctival and limbal keratinocytes were obtained from biopsies and treated with trypsin (0.05 trypsin and 0.01% EDTA) at 37 °C for about 120 min. Cells were collected every 30 min and seeded at a cell density 3–4.5 × 10^4^/cm^2^ on feeder layer (FL) of lethally irradiated 3T3-J2 cells (a gift from Prof. Howard Green) plated at the same cell density, then cultured in incubator with 5% CO_2_. The culture medium was composed of DMEM and Ham’s F12 medium (2:1 mixture) containing FBS (10%) and penicillin/streptomycin, insulin (5 μg/mL), adenine (0.18 mM), hydrocortisone (0.4 µg/mL), cholera toxin (0.1 nM), triiodothyronine (2 nM) and glutamine (4 mM). Epidermal growth factor was added at 10 ng/mL beginning at the first feeding, 3 days after plating. Subconfluent primary cultures were then passaged at a density of 6–8.3 × 10^3^ cells/cm^2^. In serial propagation assays, cells were passaged before confluence until replicative senescence.

### 4.4. Clonal Analysis and Colony-Forming Efficiency Assay

Subconfluent epithelial cultures were trypsinized, serially diluted and plated in 96-well plates (0.5 cell per well dilution) on a lethally irradiated FL of 3T3-J2 cells. After 7 days of cultivation, colonies derived from single keratinocytes were identified using an inverted microscope and trypsinized. One quarter of the colony was used for colony-forming efficiency (CFE) assay. In this assay, a small aliquot of cells was cultured for 12 days onto a 100 mm dish, then fixed and stained with rhodamine B for the classification of clonal type. This was determined by the percentage of aborted colonies formed by the progeny of the founding cell [23,24]. When 0–5% of colonies were abortive, the clone was scored as holoclone (stem cell). When more than 95% of the grown colonies were abortive (or when there were no colonies formed), the clone was classified as paraclone (terminally differentiated transient amplifying (TA) cell). Finally, when the percentage of abortive colonies was between 5% and 95%, the clone was classified as a meroclone (TA cell). The remaining three quarters of the colony were used for subculture for RNA and protein analysis.

### 4.5. Microarray Analyses

Subcultures of 32 holoclones (Table 1) from oral mucosa, limbus and conjunctival epithelia were performed.

Analysis of holoclones’ transcriptomes was carried out using Affymetrix HG-U133 Plus 2.0 array (Thermo Fisher Scientific, Waltham, MA, USA) [20]. Keratinocytes subcultured from each holoclone were feeder-depleted using immunomagnetic beads (Miltenyi Biotec, Bergisch Gladbach, Germany). According to the manufacturer’s protocol, total RNA was isolated with the Invitrogen™ PureLink™ RNA Micro Scale Kit (Thermo Fisher Scientific, Waltham, MA, USA). Differentially expressed genes (DEGs) were identified on robust multiarray average (RMA)-normalized data through the ANOVA module supplied by the Partek GS. 6.6 Software Package (ver. 7.21.1119, Chesterfield, MO, USA). The probesets displaying a fold change contrast ≥ 2 and a false discovery rate (FDR) < 0.05 were selected as DEGs among oral mucosa, limbal and conjunctival holoclones. Integral gene expression data are deposited to the Gene Expression Omnibus repository (http://www.ncbi.nlm.nih.gov/geo; series GSE198408). The network of angiogenesis-related transcripts was generated using QIAGEN IPA^®^ (ver. 8.6, QIAGEN Inc., Hilden, Germany, https://digitalinsights.qiagen.com/IPA) [80].

### 4.6. Real-Time PCR

Total RNA was isolated with the Invitrogen™ PureLink™ RNA Micro Scale Kit (Thermo Fisher), according to the manufacturer’s protocol. RNA samples were treated with RNase-free ezDNase enzyme to digest the gDNA, and SuperScript IV VILO Master Mix (Thermo Fisher) was used to synthesize the cDNA. Real-time quantitative RT-PCR was performed using TaqMan Gene Expression Assays probes (Thermo Fisher) for *PITX2* (Hs04234069_mH), complementary to all its isoforms, and *GAPDH* (Hs99999905_m1), and TaqMan Fast Advanced Master Mix (Thermo Fisher). For isoforms analysis, RT-PCR with PowerUp SYBR Green Master Mix (Applied Biosystems, Thermo Fisher) was performed using specific primers according to the literature or modified (Table 2; [25]). Reactions were run in a QuantStudio12K Flex Real Time System or in a 7900HT Fast Real-Time PCR System (Applied Biosystems, Thermo Fisher) with specific cycling programs for each master mix, as requested by manufacturer’s instructions. The expression of target genes was normalized to the level of GAPDH in the same cDNA using 2^−ΔΔCT^ quantification. For statistical analysis, Mann-Whitney test or one-way ANOVA test were applied using PRISM 8 software (version 8.4.0, GraphPad Software, San Diego, CA, USA).

### 4.7. In Situ Hybridization (ISH)

Human tissues from biopsies were formalin-fixed, paraffin-embedded (FFPE) and sectioned at 3–4 µm. *PITX2* RNA probes were hybridized to sections using the BaseScope RED Assay kit (Advanced Cell Diagnostic, Inc., Newark, CA, USA). Target retrieval was performed for 15 min with target retrieval reagent (Advanced Cell Diagnostic, Inc.) at 95 °C and for 15 min with protease III (Advanced Cell Diagnostic, Inc.) at 40 °C. The analyzed biopsies were samples of oral mucosa and ocular surface (n = 3). Probes for peptidyl-prolyl-cis-trans isomerase B and Bacillus subtilis DapB genes were used as positive and negative controls, respectively (Advanced Cell Diagnostic, Inc.).

### 4.8. Immunofluorescence and Immunohistochemistry

For ex vivo immunofluorescence (IF) studies, human tissues were embedded in an optimal cutting temperature compound (OCT) (Killik; Bio-Optica, Milan, Italy), frozen and cut into 5–7 µm sections on a cryostat (Leica 1850 UV, Wetzlar, Germany). Subsequently, sections were fixed for 10 min with 3% PFA at room temperature (RT). Then, samples were permeabilized with 0.2% Triton X-100 in PBS (20 min, RT), treated in the dark with 3% H_2_O_2_ (5 min, RT) and blocked with 2% BSA-5% FBS- 0.1% Triton X-100 (30 min, 37 °C). After the samples’ incubation for 1 h at 37 °C with the primary polyclonal antibody anti-PITX2 (ab98297, 1:1.000, Abcam, Cambridge, UK) and thereafter with the secondary antibody anti-rabbit Alexa Fluor 488 (A-21206, Thermo Fisher) (30 min, 37 °C, 1:200), the nuclei were labeled with DAPI (3 min, RT), and slides were mounted with Fluorescent Mounting Medium (Dako, Agilent Technologies, Santa Clara, CA, USA). Washes after primary and secondary antibodies were performed with 0.2% BSA, the others with 1X PBS.

For immunohistochemical (IHC) analysis, human biopsy tissues were formalin-fixed, paraffin-embedded (FFPE) and sectioned at 3–4 µm, and immunostaining was performed on the automated system Ventana BenchMark XT (Roche, Basel, Switzerland) with the primary monoclonal antibody anti-PITX2 (Abcam, ab55599, 1:100, 60 min) using diaminobenzidine as chromogen for the Ventana Ultraview Universal DAB Detection kit (Roche, Basel, Switzerland). The FFPE sections were deparaffinized and antigen retrieval was performed through the Ventana Cell Conditioning 1 antigen retrieval buffer (extended, 90 min) (Roche, Basel, Switzerland). Then, slides were counterstained with Ventana Haematoxylin II (Roche, Basel, Switzerland). Immunohistochemical sections were acquired using Imager.M2 microscope (Zeiss, Oberkochen, Germany) and AxioVision SE64 software (Rel. 4.9.1, Zeiss, Oberkochen, Germany). Measure of positive epithelium’s length for each marker was performed using MosaiX and Length software tools. Average and standard deviation (n = 3) of each marker at each follow-up timepoint was calculated. For statistical analysis, two-way ANOVA test was performed using PRISM 8 software (version 8.4.0, GraphPad Software, San Diego, CA, USA).

## 5. Conclusions

In conclusion, a microarray analysis of holoclones derived from the cornea, conjunctiva and oral mucosa tissue highlighted the pro- and antiangiogenic factors differentially expressed in these tissues, providing insights into the mechanisms involved in ocular surface neovascularization. Moreover, we identified and validated PITX2 as a new marker that univocally identifies the oral mucosa epithelium on an ocular surface regenerated through the COMET procedure. This novel marker was employed to analyze the corneal buttons of an LSCD patient followed over 10 years after COMET treatment, demonstrating the long-term stability and regeneration of oral mucosa tissue. Altogether, these findings sustained the “engraftment” hypothesis, although leaving open the possibility of the stimulation of residual LSC too. These insights will be helpful in future investigations decrypting the biological mechanisms underpinning ocular surface regeneration through oral mucosa transplantation.

## Figures and Tables

**Figure 1 ijms-24-11522-f001:**
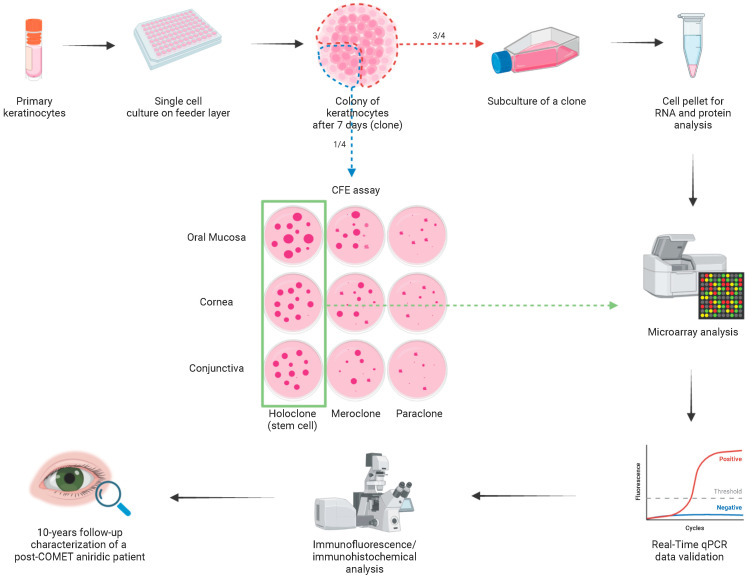
Workflow of the study. Schematic representation of the workflow related to the present study (created with Biorender.com). Abbreviations: CFE, colony-forming efficiency; COMET, cultivated oral mucosa epithelial transplantation.

**Figure 2 ijms-24-11522-f002:**
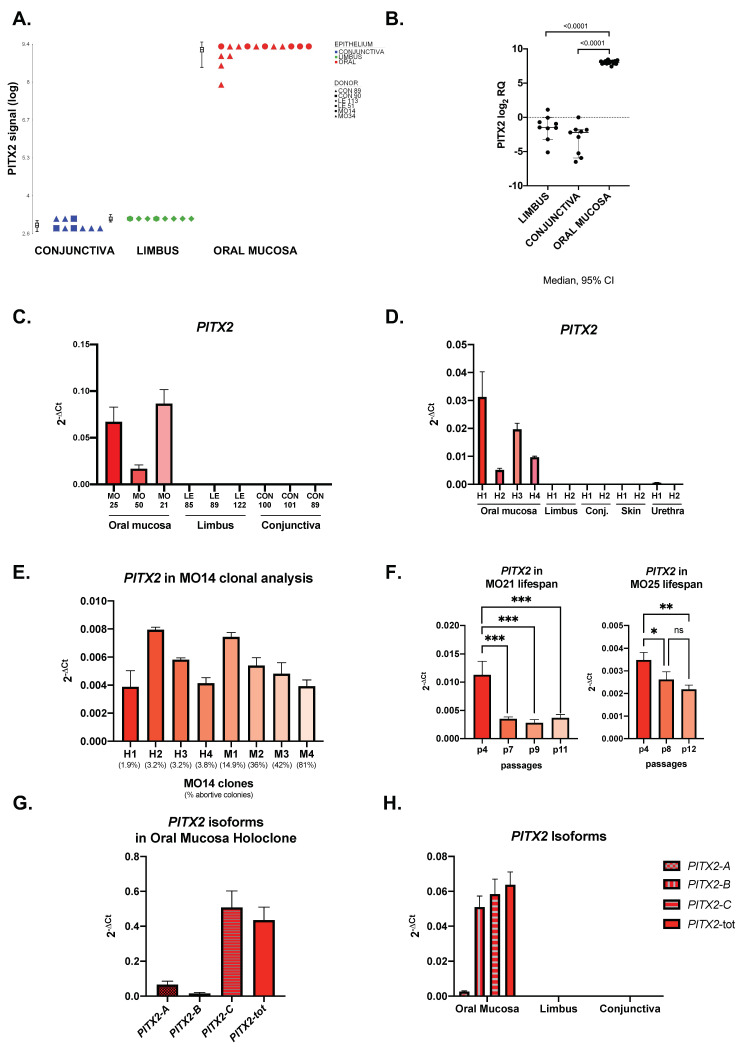
Identification and validation of *PITX2* as differentially expressed transcript: (**A**) Boxplots showing *PITX2* signals in conjunctival, limbal and oral mucosal holoclones (results are expressed in log scale); (**B**) Real-time RT-PCR validation of *PITX2* transcript levels expressed as log_2_RQ. Mann-Whitney test performed; (**C**) Real-time RT-PCR of *PITX2* in primary cultures of young passages (p3-p6) of oral mucosa, limbus (cornea) and conjunctiva; (**D**) Real-time RT-PCR of *PITX2* expression in holoclones of oral mucosa, limbus (cornea), conjunctiva, skin and urethra; (**E**) Real-time RT-PCR of *PITX2* in holoclones and meroclones of a strain of oral mucosa ordered by increasing % of abortive colonies (decreasing clonogenicity); (**F**) Real-time RT-PCR of *PITX2* in passages of lifespans of two different donors of oral mucosa. ns = not significant, *** = *p* < 0.0004, ** = *p* < 0.005, * = *p* < 0.03 with one-way ANOVA test; (**G**) Real-time RT-PCR of *PITX2* isoforms in a holoclone of oral mucosa; (**H**) Real-time RT-PCR of *PITX2* isoforms in primary cultures of young passages of oral mucosa, cornea and conjunctiva. Abbreviations: CON, conjunctiva; LE, limbus; MO, oral mucosa; H, holoclone; M, meroclone; *PITX2*-tot, all *PITX2* isoforms.

**Figure 3 ijms-24-11522-f003:**
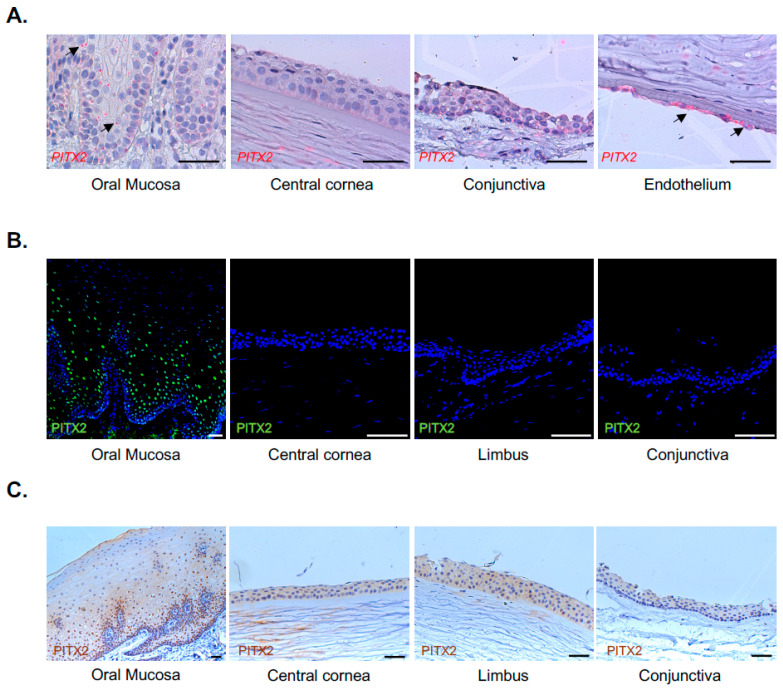
Validation in PITX2 expression in vivo: (**A**) In situ hybridization of *PITX2* transcript (red) on formalin-fixed paraffin-embedded (FFPE) biopsies of oral mucosa, central cornea, conjunctiva and corneal endothelium (n = 3). Black arrows highlight positive signals; (**B**) In vivo immunofluorescence analysis of PITX2 protein (green) in samples of oral mucosa (n = 7), central cornea, limbus and conjunctiva (n = 3); (**C**) Immunohistochemical analysis of PITX2 protein (brown) on formalin-fixed paraffin-embedded biopsies of oral mucosa, central cornea, limbus and conjunctiva (n = 3). Scale bars = 50 μm.

**Figure 4 ijms-24-11522-f004:**
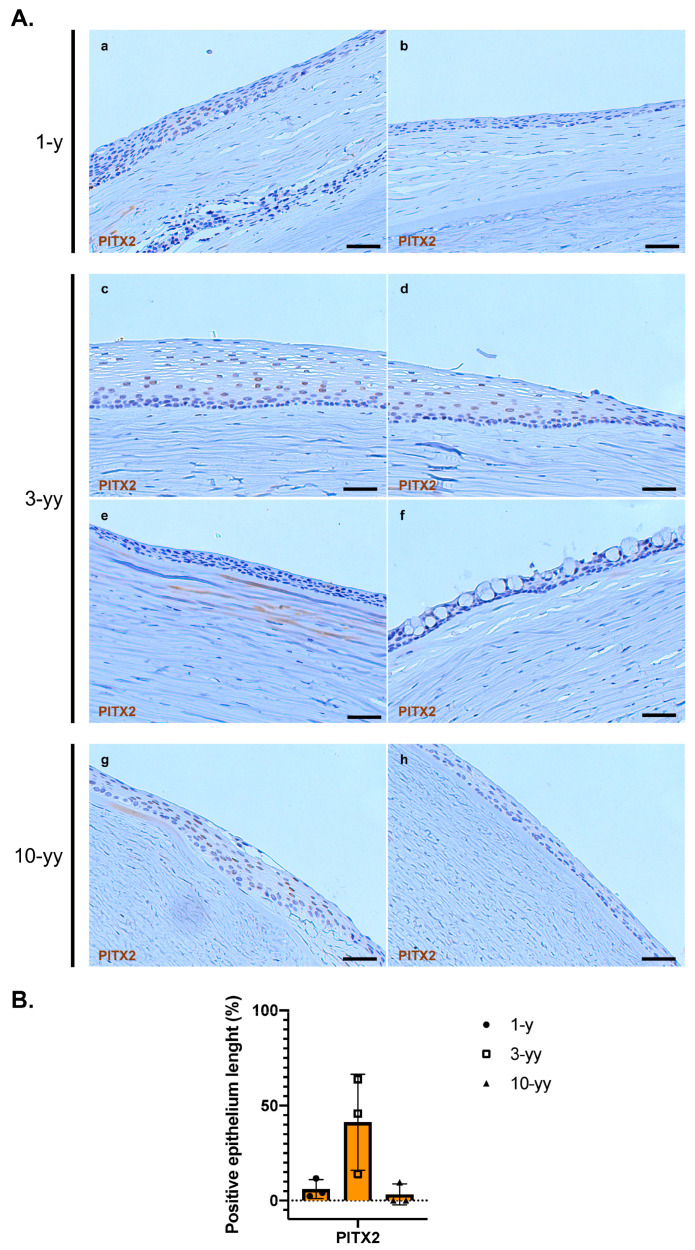
Phenotypic characterization of the corneal buttons derived from the PKs of a patient treated by COMET: (**A**) Immunohistochemical analysis of the corneal buttons from the PKs at 1 (**a**,**b**), 3 (**c**–**f**) and 10 years (**g**,**h**) post-COMET. Representative images of PITX2-positive (**a**,**c**,**d**,**g**) and -negative (**b**,**e**,**f**,**h**) areas are shown in the panel; the images related to the 3-year corneal button highlighted a transitional zone between oral mucosa and another epithelium (**d**) and a zone with conjunctival epithelium with a cluster of goblet cells (**f**). Scale bars = 50 μm. (**B**) Graph showing the percentage of the positive epithelial length for PITX2 in the three PK corneal buttons (n = 3). Abbreviations: PK, penetrating keratoplasty; y/yy, year/s.

**Figure 5 ijms-24-11522-f005:**
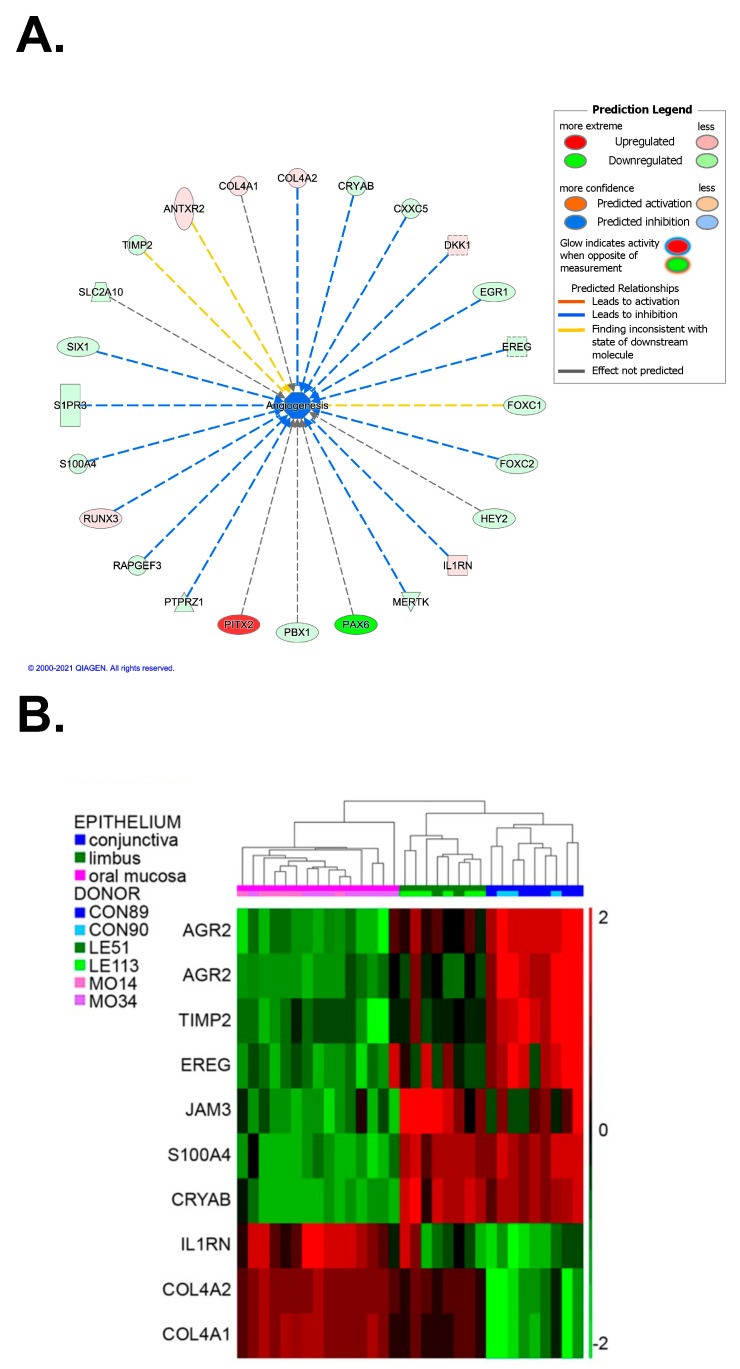
Analysis of angiogenesis-related factors in microarray comparisons: (**A**) Ingenuity Pathway Analysis of the DEGs in oral mucosa versus conjunctiva related to angiogenesis (further details concerning the iconography are available at https://qiagen.my.salesforce-sites.com/KnowledgeBase/articles/Knowledge/Legend (accessed on 14 July 2023)); (**B**) heatmap describing gene expression profiles of DEGs in oral mucosa in comparison with limbus and conjunctiva related to angiogenesis (created using Partek^®^ software). Abbreviations: CON, conjunctiva; LE, limbus; MO, oral mucosa. The ANOVA-based pairwise comparison among the three epithelia highlighted the differentially expressed genes (DEGs, fold change (FC) ≥ 2 and FDR < 0.05 for upregulated; FC ≤ −2 and FDR < 0.05 for downregulated) in oral mucosa vs. cornea or vs. conjunctiva (Spreadsheets S1–S3 in [20]).

**Figure 6 ijms-24-11522-f006:**
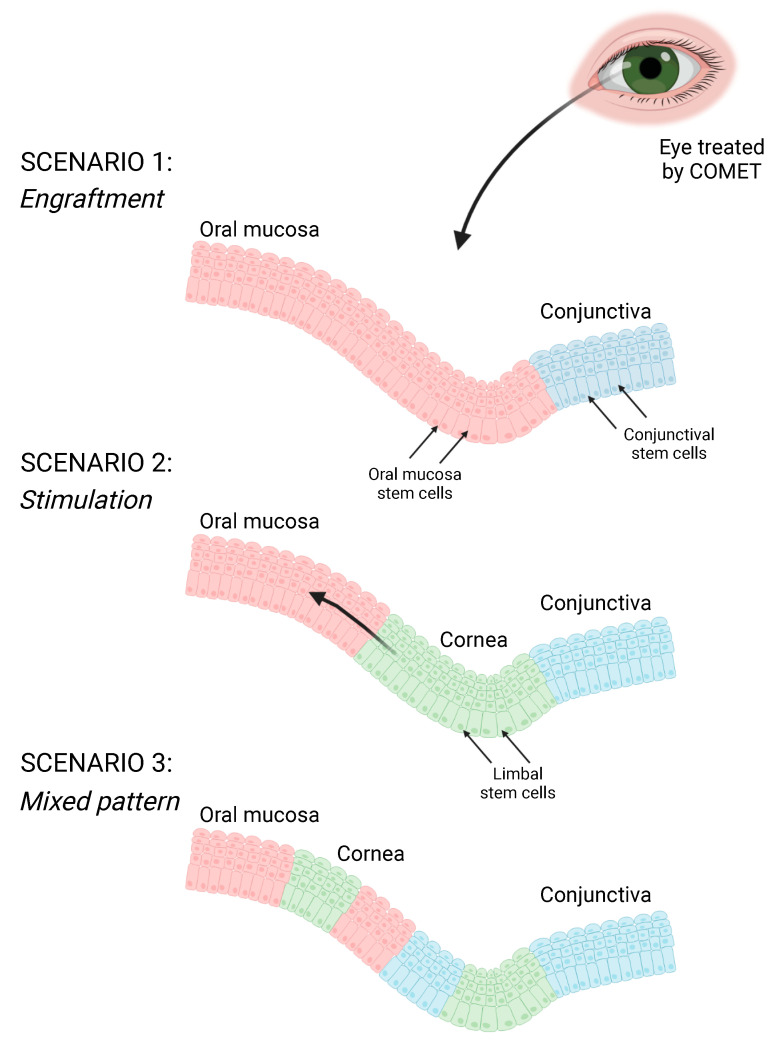
Possible scenarios of the mechanisms of ocular surface regeneration after COMET procedure (created with Biorender.com).

**Table 1 ijms-24-11522-t001:** Strains and their holoclones analyzed. Abbreviations: CON: conjunctiva; LE: limbus; MO: oral mucosa.

Epithelium	Strain	N. of Holoclones
Conjunctiva	CON-89	6
	CON-90	3
Limbus	LE-51	2
	LE-113	6
Oral Mucosa	MO-14	6
	MO-34	9

**Table 2 ijms-24-11522-t002:** List of forward and reverse primers used for isoform analysis of *PITX2* [25].

Target Gene	Forward Primer (5′-3′)	Reverse Primer (5′-3′)
*PITX2* tot	CAGCCTGAGACTGAAAGCA	GCCCACGACCTTCTAGCAT
*PITX2A*	GCGTGTGTGCAATTAGAGAAAG	CCGAAGCCATTCTTGCATAG
*PITX2B*	GCCGTTGAATGTCTCTTCTC	CCTTTGCCGCTTCTTCTTAG
*PITX2C*	ACTTTCCGTCTCCGGACTTT	CGCGACGCTCTACTAGTC
*GAPDH*	GACCACAGTCCATGCCATCAC	TCCACCACCCTGTTGCTGTAG

## Data Availability

Integral gene expression data are deposited to the Gene Expression Omnibus repository (http://www.ncbi.nlm.nih.gov/geo; series GSE198408).

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
