# Peer review of "Comparison between Cultivated Oral Mucosa and Ocular Surface Epithelia for COMET Patients Follow-Up"

_ijms, 2023, doi:10.3390/ijms241411522_

Round 1
Reviewer 1 Report
Attico et al reported the resuts compared between culticated oral mucosa and ocular surface epithelia for COMET patients follow-up. This research is of high quality. However, minor errors are found. Please correct them.
Abstract, line 23,
PITX2 is an abbreviation, so please provide the full spelling.
1.Introduction
Line 62: GFP is an abbraviation. provide the full spelling.
Line 350: The line break on line 350 appears to be unnecessary.
Line 354: As the abbreviation PITX2 has already been introduced, its full spelling is not necessary here.
Figure 4: In the caption, more explanation is needed regarding parts a through h. Please add a description of the differences between a and b, c and d, and g and h. Also, it is unclear from where the tissue in e was obtained.
Figure 5: The predictive legend is too small and difficult to discern. Please enlarge it for better readability.
Caption for figure 5, line 224: Please change the "?" character to superscript.
Refferences: The information for many of the cited sources is incomplete. Please correct the incomplete portions.
Ref numbers: 9, 11, 16, 20, 32, 32, 34, 38, 51, 63, 67, 71,74, 77, 78.
Reviewer 2 Report
To better characterize oral mucosal cell-derived corneal transplants, the authors compared the microarray transcriptome profile of human oral mucosa, limbal and conjunctival holoclones. They identified PITX2 as a new marker distinguishing the transplanted oral tissue from the other epithelia. They further validated PITX2 at RNA and protein levels to investigate 10-year follow-up corneal samples derived from a COMET-treated aniridic patient. They found novel angiogenesis-related factors differentially expressed in the three epithelia and important in explaining the neovascularization in COMET-treated patients. These novel results lay the foundation for the follow-up analysis of patients transplanted with oral mucosa and uncover new aspects of the regeneration mechanism of the transplanted corneas.
I have only minor concerns about this excellent manuscript.
1. Please check the picture quality. It appears that the resolution is too low from uploading non-pdf pictures.
2. In the results, as they precede the methods, please briefly define holoclones because this is a general journal. In the abstract, the authors might like to substitute "cells" to "cultured holoclones" when mentioning transcriptome analysis. Also, please add "gene array" to "transcriptomic analysis".
3. In Fig. 2, please consider more straightforward abbreviations for an easier understanding.
4. The human genes/transcripts should be presented in italics and in uppercase (https://www.biosciencewriters.com/Guidelines-for-Formatting-Gene-and-Protein-Names.aspx#:~:text=Gene%20symbols%20are%20italicized.,Protein%20symbols%20are%20not%20italicized). Proteins may be presented in lowercase.
5. The notion about ColIV chains as being antiangiogenic may be a stretch here, because there are no data on their processing/cleavage in this study. Please tone down or remove.
6. TIMP2 is not a metalloproteinase but its inhibitor.
7. In Conclusions, please mention which hypothesis the present results support.
8. Please identify the institutional body permitting this work (e.g., IRB).
9. For FFPE specimens please specify whether they were deparaffinized.
10. Section 5.8. Please change "in vivo" to "ex vivo" in the first sentence.
